# Infused-liquid-switchable porous nanofibrous membranes for multiphase liquid separation

Yang Wang[1], Jiancheng Di[1], Li Wang[2], Xu Li[1], Ning Wang[1], Baixian Wang[1], Ye Tian[3,4], Lei Jiang[2,4] & Jihong Yu[1,5]

Materials with selective wettabilities are widely used for effective liquid separation in environmental protection and the chemical industry. Current liquid separation strategies are primarily based on covalent modification to control the membranes' surface energy, or are based on gating mechanisms to accurately tune the gating threshold of the transport substance. Herein, we demonstrate a simple and universal polarity-based protocol to regulate the wetting behavior of superamphiphilic porous nanofibrous membranes by infusing a high polar component of surface energy liquid into the membranes, forming a relatively stable liquid-infusion-interface to repel the immiscible low polar component of surface energy liquid. Even immiscible liquids with a surface energy difference as small as $2\,mJ\,m^{-2}$, or emulsions stabilized by emulsifiers can be effectively separated. Furthermore, the infused liquid can be substituted by another immiscible liquid with a higher polar component of surface energy, affording successive separation of multiphase liquids.

[1] State Key Laboratory of Inorganic Synthesis and Preparative Chemistry, College of Chemistry, Jilin University, Changchun 130012, China. [2] Key Laboratory of Bio-inspired Materials and Interfacial Science, Technical Institute of Physics and Chemistry, Chinese Academy of Sciences, Beijing 100190, China. [3] Beijing National Laboratory for Molecular Sciences (BNLMS), Key Laboratory of Green Printing, Institute of Chemistry, Chinese Academy of Sciences, Beijing 100190, China. [4] University of Chinese Academy of Sciences, Beijing 100049, China. [5] International Center of Future Science, Jilin University, Changchun 130012, China. Correspondence and requests for materials should be addressed to Y.T. (email: tianyely@iccas.ac.cn) or to J.Y. (email: jihong@jlu.edu.cn)

Much effort has been given to the separation of water (surface energy, SE = 72.8 mJ m$^{-2}$) and oils (mainly SEs < 35.0 mJ m$^{-2}$) via the utilization of membranes with special wetting behaviors[1–5]. However, the separation of immiscible liquids with a smaller difference in SE has been less explored, and successive separation of multiphase liquids remains a challenge. Such a separation process is, in fact, highly demanded in the chemical industry, for example, during anhydrous heterogeneous chemical reactions[6–8] and multi-liquid-phase extraction[9]. For the separation of these immiscible liquids, Jiang and his co-workers proposed that the SE of the membrane should be controlled in the middle of the intrinsic wetting thresholds of the immiscible liquids[10]. Such membranes could both allow the flow of low SE liquid and block the flow of the high SE liquid. However, this approach suffers from inherent limitations that severely restrict its applicability. For example, it is relatively complicated to manipulate the SEs of membranes via covalent modification[10–12], especially for the separation of immiscible liquids with a small SE difference.

Recently, a type of liquid-infused microtextured membranes inspired by the *Nepenthes* pitcher plant[13] has been developed for liquid separation, in which the infused immiscible liquid layer acts as the repellent surface rather than the covalently modified solid surface. The premise for these membranes is that the SE of the infused liquid should match that of the textured substrate and form a stable state. Aizenberg and co-workers reported that the separation of a three-phase air–water–oil mixture could be realized by accurately tuning the gating threshold[14]. However, during the separation process, extra pressure was needed to open the pores sealed by the lubricating liquid to permit the flow of liquid. The extra pressure must be precisely adjusted according to different gating thresholds of the transport substance.

Herein, we develop a universal infused-liquid-switchable separation protocol based on polarity interaction between liquids and membranes, without any external stimulation or covalent modification. Flexible SiO$_2$–TiO$_2$ composite porous nanofibrous membranes (STPNMs) are used to demonstrate this protocol for the separation of arbitrary immiscible liquids, as well as for successive multiphase liquid separation. As reported by Fowkes[15, 16], the SE is comprised of a dispersive part of SE (DSE) and a polar part of SE (PSE) (Supplementary Table 1), and the polar and dispersive interfacial attraction can be treated independently. In our system, the PSE of the membrane is much larger than DSE ($\gamma_S^p \gg \gamma_S^d$), thus the polar-polar interaction is dominant. We use the PSE as the functional parameter instead of the total SE. The wetting behavior of the superamphiphilic STPNMs can be regulated by infusing a high PSE liquid into the membranes, forming a relatively stable liquid-infusion-interface (LII) to repel the immiscible low PSE liquid. Even multiphase mixtures of immiscible liquids with a SE difference as small as 2 mJ m$^{-2}$, or emulsions stabilized by emulsifiers can be effectively separated, without any covalent modification to control the SE of membranes.

## Results

**Morphologies and properties of the STPNMS.** As illustrated in Supplementary Fig. 1, STPNMs are prepared by electrospinning a viscous precursor solution through a syringe under a high voltage electric field[17]. These composite fibers possess excellent flexibility with appropriate titanium doping concentration (Supplementary Fig. 2). The scanning electron microscopy (SEM) image in Fig. 1a reveals that the STPNMs are composed of entangled uniform fibers with a diameter of 200–300 nm. The transmission electron microscopy (TEM) images of STPNMs exhibit the worm-like mesoporous structure with random orientations (Fig. 1b, c). Nitrogen adsorption–desorption isotherms of STPNMs in Supplementary Fig. 3a correspond to typical IV-like isotherms, proving the existence of mesopores in STPNMs[18]. The Brunauer–Emmett–Teller surface area of STPNMs reaches 652.9 m$^2$ g$^{-1}$ and the pore volume of mesopores is 0.33 cm$^3$ g$^{-1}$. Calculated by nonlocal density functional theory on the adsorption branch, the pore sizes are mainly distributed between 1.2 and 2.0 nm (Supplementary Fig. 3b). The Fourier transform infrared spectrum (Supplementary Fig. 4) clearly reveals the broad peak at 3424 cm$^{-1}$, indicating the presence of abundant Si–OH and Ti–OH groups[17, 19]. SiO$_2$–TiO$_2$ composite nanofibrous membranes (STNMs) with the same chemical composition to STPNMs, but without adding surfactant are also fabricated by electrospinning technique (Supplementary Fig. 5). STNMs possess much smaller surface area (155.1 m$^2$ g$^{-1}$) and the pore sizes distribute over a wide range from 0.5 to 20 nm (Supplementary Fig. 3). Supplementary Fig. 6 shows the tensile stress vs. strain curves of STPNMs and STNMs. The derived Young's modulus and stress of break of STPNMs are much larger than those of STNMs.

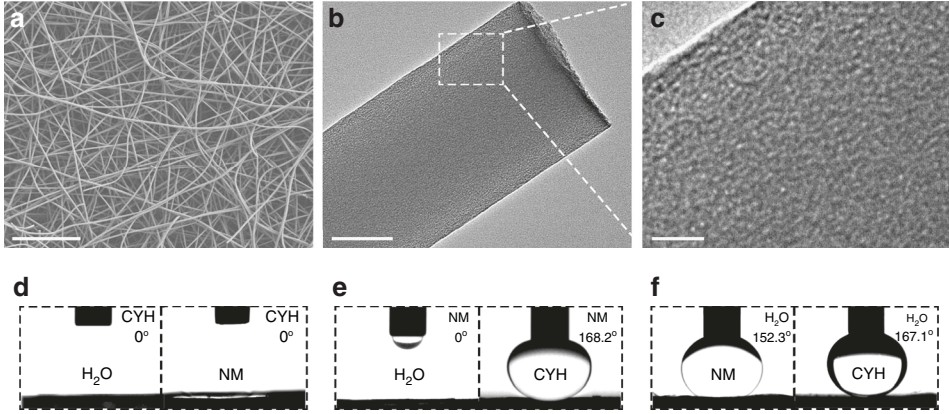

**Fig. 1** Nanostructure and infused-liquid-switchable wetting behavior of STPNMs. **a** SEM image of STPNMs, showing the entangled uniform fibers with a diameter of 200–300 nm. **b, c** TEM images of STPNMs exhibit the worm-like mesopores with orientational randomization in the fibers. *Scale bars*, **a**: 10 μm; **b**: 100 nm; **c**: 20 nm. **d–f** The switchable wetting behavior of STPNMs when immersed in cyclohexane, nitromethane and water, respectively, and named as liquid-LII. 1 μl of liquid droplet is used as indicator for all of the liquids. **d** The CYH-LII is lyophilic for water and NM. **e** The NM-LII is lyophilic for water but lyophobic for CYH. **f** The water-LII is lyophobic for both NM and CYH

**Table 1 Infused-liquid-switchable wetting behavior of STPNMs for a series of liquid pairs**

| SE (mJ m$^{-2}$) | 72.8 | 58 | 48.8 | 36.5 | 44 | 36.8 | 33.3 | 28.4 | 50.8 | 27 | 25.24 | 25.22 | 20.25 | 18.4 |
|---|---|---|---|---|---|---|---|---|---|---|---|---|---|---|
| PSE (mJ m$^{-2}$) | 51 | 19 | 16 | 11.3 | 8 | 7 | 2.5 | 2.3 | 1.8 | 0.3 | 0 | 0 | 0 | 0 |
| RL \ IL | H$_2$O | FM | EG | DMF | DMSO | NM | ED | TL | DIM | CCl$_4$ | CYH | KS | PE | NM |
| NH | + | + | + | + | + | + | ● | ● | ● | ● | ● | ● | ● | ● |
| PE | + | + | + | + | + | + | ● | ● | ● | ● | ● | ● | ● | ● |
| KS | + | + | + | + | + | + | ● | ● | ● | ● | ● | ● | ● | ● |
| CYH | + | + | + | + | + | + | ● | ● | ● | ● | ● | ● | ● | ● |
| CCl$_4$ | + | + | + | ● | ● | ● | ● | ● | ● | ● | ● | ● | ● | ● |
| DIM | + | + | + | ● | ● | + | ● | ● | ● | ● | ● | ● | ● | ● |
| TL | + | + | + | ● | ● | ● | ● | ● | ● | ● | ● | ● | ● | ● |
| ED | + | + | + | ● | ● | ● | ● | ● | ● | ● | – | – | – | – |
| NM | + | ● | ● | ● | ● | ● | ● | ● | ● | ● | – | – | – | – |
| DMSO | ● | ● | ● | ● | ● | ● | ● | ● | ● | ● | – | – | – | – |
| DMF | ● | ● | ● | ● | ● | ● | ● | ● | ● | ● | – | – | – | – |
| EG | ● | ● | ● | ● | ● | – | – | – | – | – | – | – | – | – |
| FM | ● | ● | ● | ● | ● | ● | – | – | – | – | – | – | – | – |
| H$_2$O | ● | ● | ● | ● | ● | – | – | – | – | – | – | – | – | – |

+ lyophobic, -: lyophilic, ●: miscible, CCl$_4$ tetrachloromethane, CYH cyclohexane, DIM diiodomethane, DMF N, N'-dimethylformamide, DMSO dimethylsulfoxide, ED ethane dichloride, EG ethylene glycol, FM formamide, IL infused liquid, KS kerosene, LIM liquid-infused membrane, NH n-hexane, NM nitromethane, PE petroleum ether, RL repellent liquid, TL toluene. The liquids are arranged according to the PSEs from high to low. The high PSE liquids-infused STPNMs show lyophobic to the immiscible low PSE liquids; while the low PSE liquids-infused STPNMs show lyophilic to the immiscible high PSE liquids

**Infused-liquid-switchable wetting behavior of STPNMs.** Three immiscible liquids with different PSEs, including water (51 mJ m$^{-2}$), nitromethane (NM, 7 mJ m$^{-2}$), and cyclohexane (CYH, 0 mJ m$^{-2}$) are chosen as examples to demonstrate the infused-liquid-switchable wetting behavior of STPNMs based on polarity. As shown in Supplementary Fig. 7, the pristine STPNMs are superamphiphilic in air for all of the three liquids. However, the wetting behavior of the LII formed by infusing different liquids into STPNMs is switchable: the CYH-LII is lyophilic for water and NM; the NM-LII is lyophilic for water but lyophobic for CYH; the water-LII is lyophobic for both NM and CYH (Fig. 1d–f). To prove the interchangeability of liquids absorbed in STPNMs[20], a non-volatile perfluorinated lubricant liquid (Fluorinert FC-43) used as the low PSE liquid, is infused into STPNMs. After being washed with water, the characteristic peak of F 1 s (Supplementary Fig. 8) on STPNMs completely disappears, ascertaining thorough substitution of FC-43 by water.

**Generality in separation of liquid mixtures.** Table 1 lists the wetting behavior of STPNMs for a series of liquid pairs with different PSE. It can be seen that the wetting behavior of STPNMs is universal for arbitrary immiscible liquids: the high PSE liquid-LII shows lyophobicity to the low PSE liquids, while low PSE liquid-LII is lyophilic to the high PSE liquids and can be transformed into high PSE liquid-LII. Utilizing this polarity-based protocol, even immiscible liquids with a PSE difference as small as 5.2 mJ m$^{-2}$ (NM and diiodomethane) can be effectively

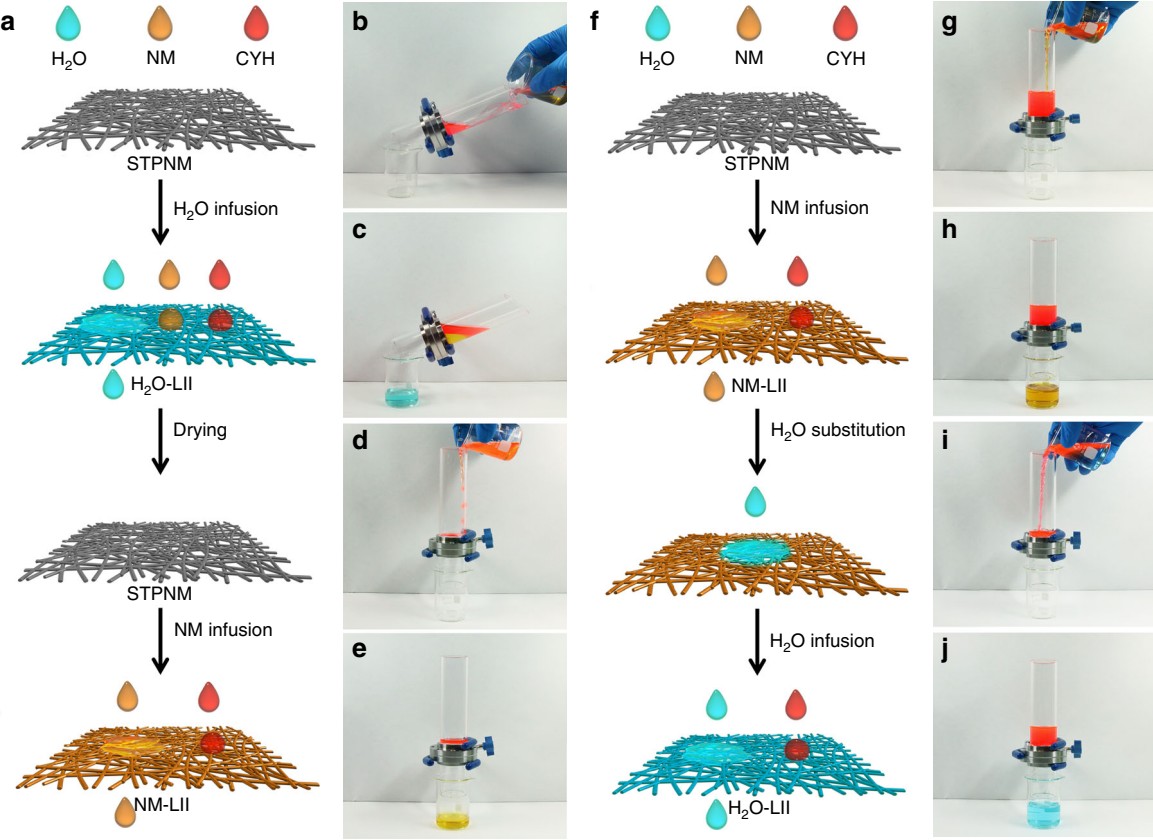

**Fig. 2** Separation of multiphase liquids. **a** Schematic illustration of multiphase liquids separation (water, CYH and NM). Water selectively permeates through the water-LII, while NM and CYH are retained. After that, the water-LII is dried and then infused by NM to form NM-LII. The NM-LII allows the passage of NM, but retains CYH. (**b–e**) Demonstration of the separation process of the multiphase liquids. **b** The mixture of water, NM and CYH (volume ratio 1:1:1) is poured onto a water-infused STPNM. **c** The separated water is collected and no other colorful liquids are observed. **d**, **e** After the drying and NM infusion process, the separation of CHY and NM is achieved by NM-LII membrane and no red liquid is visible in the collector. **f** Schematic illustration of successive separation of two pairs of immiscible liquids (NM/CYH and water/CYH) over one STPNM. NM selectively permeates through the NM-LII, while CYH is blocked. Then, a small quantity of water is added to substitute NM in the NM-LII, making it into water-LII. The water-LII allows the passage of water, but retains CYH. **g–j** Presentation of the successive separation process. **g**, **h** Mixture of CYH and NM (volume ratio 1:1) is separated by NM-infused STPNM and only yellow liquid can be observed in the beaker. **i**, **j** After the adding of a small quantity of water, mixture of water and CYH (volume ratio 1:1) is separated by water-LII, and no red liquid is visible in the collector

separated (Supplementary Fig. 9a). Notably, we can achieve the challenging separation of ethyl glycol and diiodomethane with a SE difference of only $2 \, mJ \, m^{-2}$ based on their PSE difference ($14.2 \, mJ \, m^{-2}$) (Supplementary Fig. 9b).

To further evaluate the separation capability of STPNMs, two emulsions are prepared (Supplementary Fig. 10): surfactant-stabilized oil-in-water (CYH/water, stabilized by CTAB) and oil-in-oil (CYH/formamide, stabilized by Pluronic F-127). For both emulsions, densely-packed droplets are completely removed after separation, indicating the effectiveness of the STPNMs for separating various emulsions.

**Successive separation of multiphase liquids**. Figure 2a–e presents the multiphase liquids separation process. For the mixture of these liquids, water selectively permeates through water-LII while the mixture of NM and CYH is retained. After that, the separation membrane is heated to remove the adsorbed water and then infused with NM to form NM-LII. Along with the selective permeation of NM, the separation of residual NM and CYH is achieved by NM-LII. It can be seen that the mixture is effectively separated and no immiscible liquid is visible.

As the infused liquid can be directly substituted by another high PSE liquid, we have realized the successive separation of two

pairs of immiscible liquids (NM/CYH and water/CYH). Fig. 2f–j present the separation process: First, the mixture of NM and CYH is poured onto the NM-LII. NM selectively permeates through the membrane, while CYH is intercepted and kept in the upper tube. After that, a small amount of water is added to directly substitute the adsorbed NM, making the NM-LII into water-LII. As shown in Supplementary Fig. 11, a water drop spreads out quickly once it contacts with the NM-LII and infiltrates into the membrane within 1 s. The time evolution of the spreading area of a water drop indicates that the wetting area spreads fast at the first 0.7 s and reaches the maximum area at 1.04 s. Finally, water (mixture of water/CYH) selectively permeates through the water-LII, whereas CYH is retained above the membrane. No visible immiscible liquid is observed in filtrates.

**Stability and recyclability of STPNMs**. The chemical stability of the STPNMs is evaluated by characterizing their resistance to the corrosion of salt, acid, and an organic solvent. Supplementary Fig. 12a shows the underwater contact angles (CAs) of $CCl_4$ droplets on the STPNMs after being immersed in NaCl (10 wt %), HCl (0.1 M) aqueous solutions and DMF for 20 days, respectively. The STPNMs retain their underwater superlyophobic property and no obvious change of CAs is observed after corrosive

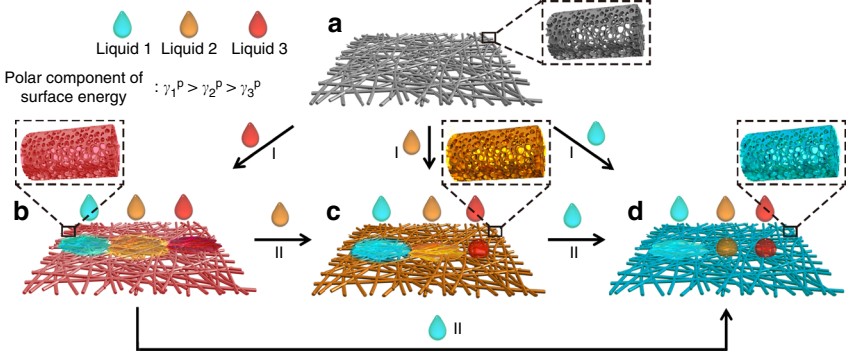

**Fig. 3** Schematic of the infused-liquid-switchable wetting behavior of STPNMs based on polarity. **a** The STPNMs are superamphiphilic in air for any immiscible liquids with different PSEs ($\gamma_1^p > \gamma_2^p > \gamma_3^p$) and can be infused to form liquid-LII, respectively (process I). The liquid-LII shows switchable wetting behavior upon the infusion by different liquids. **b** The liquid 3-LII with the lowest PSE is lyophilic for all of the three liquids. **c** The liquid 2-LII with medium PSE is lyophobic for liquid 3 but lyophilic for liquid 1. **d** The liquid 1-LII with the highest PSE is lyophobic for both liquids 2 and 3. Moreover, the infused-liquid in STPNMs can be directly substituted by immiscible liquids with a higher PSE (process II). Liquid 3-LII can be converted to liquid 2-LII once liquid 2 is added. And liquid 1 can infuse into liquid 2 and 3-LII, respectively, forming liquid 1-LII

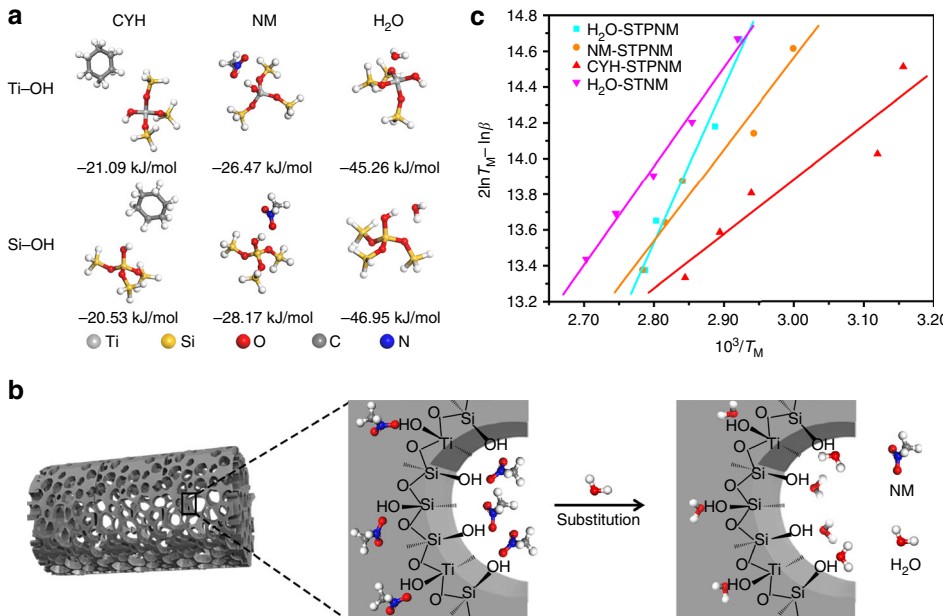

**Fig. 4** Mechanism of the wetting behavior of STPNMs. **a** Optimized geometry of guest molecules binding on surface hydroxyl groups of STPNMs by the DFT calculations and the corresponding binding energy. **b** Schematic illustration of the substitution of NM molecules by water molecules on STPNMs. **c** Plots of $(2\ln T_M - \ln\beta)$ against $10^3/T_M$ for TPD of water, NM and CYH on STPNMs and water on STNMs. The *dots* are the experimental data measured at different temperature ramping rates and the *lines* are curve-fitting results. The desorption activation energies $E_d$ are obtained by calculating the slope of the fitted curves

immersion, which proves the excellent chemical resistance ability of STPNMs.

The separation process of CYH/water or CYH/formamide (30 ml with volume ratio of 1:1) on one STPNM is repeated ten times to test the recyclability. After each separation process, the membrane is treated by heating at 200 °C for 1 h to remove the adsorbed liquid and then reused. The separation efficiency is determined by using the infrared spectrometer oil content analyzer. As shown in Supplementary Fig. 12b, the separation efficiency is above 99.9% and there is no obvious attenuation after ten cycles, which indicates good recyclability.

## Discussion

Figure 3 illustrates the infused-liquid-switchable wetting behavior of STPNMs based on polarity. The STPNMs are superamphiphilic in air, but their wetting behavior can be manipulated by the infused liquid (process I). The STPNMs preferentially capture and interact with the high PSE molecules via abundant surface hydroxyl groups, forming a relatively stable LII. The LII allows the permeation of the infused liquid itself, but repels the immiscible liquid with a lower PSE. Furthermore, the infused liquid can be directly substituted by another immiscible liquid with a higher PSE (process II), thus realizing successive separation of multiphase liquids. To understand the infused-liquid switchable behavior of the membrane, we compare the total interfacial energies of STPNMs that are completely wetted by a relatively low PSE liquid ($E_A$), and a high PSE liquid-infused membrane ($E_B$) (Supplementary Note 1)[21–24]. To ensure the solid preferentially interacts with the higher PSE liquid, $\Delta E = E_A - E_B$ must be greater than zero. The

equation can be described as:

$$\Delta E = R(\gamma_B \cos\theta_B - \gamma_A \cos\theta_A) + \gamma_A - \gamma_B \qquad (1)$$

where $\gamma_A$ and $\gamma_B$ are the surface tensions for the liquid to be repelled and the infused liquid (Supplementary Table 1), respectively, $\theta_A$ and $\theta_B$ are the equilibrium CAs of the repelled liquid and the infused liquid on a flat solid surface (Supplementary Fig. 13 and Supplementary Table 2), respectively, and $R$ is the roughness factor. The calculation results in Supplementary Table 3 agree well with the experimental results, which shows that lower PSE liquid in STPNMs can be substituted by a liquid with a higher PSE.

We further explore the wetting behavior of STPNMs by measuring the magnitude of the solid-liquid interaction force. The interaction of solid and liquid is the sum of the interface forces due to the various types of molecular attraction containing both polar and dispersive parts[25]. The polar and dispersive interfacial attractions, proposed by Fowkes[16], can be treated independently, and the polar-dispersive interactions can be neglected. The components of the surface tension of the STPNMs are estimated based on the contact angle data and fitted using the OWRK method[25, 26] (Supplementary Fig. 14 and Supplementary Note 2): $\gamma_S^d = 18.18$ mN m$^{-1}$ and $\gamma_S^p = 40.93$ mN m$^{-1}$, where the $\gamma_S^d$ and $\gamma_S^p$ correspond to dispersive and polar parts of the surface tension, respectively. As $\gamma_S^p \gg \gamma_S^d$, we replace the energy argument of the total SE in Eq. 1 with the PSE to investigate the influence of the PSE of the liquid on the stability of the liquid-LII (Supplementary Equation 3), where $\gamma$ in the original expression is replaced by $\gamma^p$. The theoretical results in Supplementary Table 3 are in accord with the experimental results, proving that the interaction force between the liquid and the STPNMs is governed by the polar-polar interaction.

Density functional theory has been employed to quantitatively calculate the binding energies between liquid molecules and polar hydroxyl groups on the STPNMs surface, and the results are presented in Fig. 4a. The binding energy of hydroxyl groups with the water molecule is much higher than that with the NM and CYH molecules, indicating that the water-LII is more stable to repel the low PSE liquids. The CYH molecule has the weakest binding energy, which means that the CYH-LII is not stable enough and will easily evolve into more stable liquid-LII upon infusion by a high PSE liquid. As the binding energy of NM is higher than that of CYH but lower than that of water, the NM-LII is stable enough to both repel CYH and be infused by water (Fig. 4b).

Temperature-programmed desorption (TPD) is used to further characterize the affinity between the liquid and STPNMs or STNMs. Typical desorption curves of water, NM and CYH on STPNMs and water on STNMs with a temperature ramping rate of 5 °C min$^{-1}$ are presented in Supplementary Fig. 15. The desorption activation energy of LII is calculated by the model proposed by Cvetanovic and Amenomiya[27] (Eq. 2):

$$2\ln T_M - \ln\beta = E_d/RT_M + \ln(E_d/AR), \qquad (2)$$

where $T_M$ is the temperature of desorption peak, $\beta$ is the temperature ramping rate, $E_d$ is the desorption activation energy, $A$ is the preexponential factor and $R$ is the gas constant. According to the slopes of plots of $(2\ln T_M - \ln\beta)$ against $10^3/T_M$ in Fig. 4c, the $E_d$ is calculated as 72.35, 42.52 and 26.60 kJ mol$^{-1}$ for water, NM and CYH on STPNMs, respectively, and 45.89 kJ mol$^{-1}$ for water on STNMs. The higher $E_d$ indicates more stable LII in dynamics, which is in accord with the experimental results. By comparing the value of $E_d$ (water) on STNM and STPNMs, the existence of mesopores not only increases the mechanical strength of the membranes, but also offers more polar groups and

higher adsorption capacity, which will enhance the stability of liquid-LII.

In summary, we have developed a simple, effective, and universal polarity-based strategy for the separation of immiscible liquids with a very low SE difference, or emulsions stabilized by emulsifiers. The STPNMs preferentially capture and interact with the high PSE molecules, which are repellent to the immiscible low PSE liquid. Furthermore, the infused liquid can be directly substituted with another immiscible liquid with a higher PSE and form a more stable LII, leading to the successful successive separation of multiphase liquids. Nevertheless, the impact of the pore size on the polarity-driven mechanism needs to be further investigated. Different from other reported liquid-infused system[22, 28], our work provides a polarity-based separation strategy, which will open new perspectives for the applications of liquid separation to environmental protection and chemical industry. However, it should be pointed out that the separation of miscible liquids (e.g., ethanol and water) with such superwetting membranes remains challenging.

## Methods

**Fabrication of STPNMs.** A total of 5.0 g of titanium (III) chloride (20 wt%, in 3% HCl aqueous solution), 7.3 g of ethanol, 100 μl of HCl (aq., 37.0 wt%) and 0.4 g of polyethylene oxide (M$_w$: 1,000,000) were mixed with continuous stirring at room temperature for 1.5 h. Subsequently, 3.75 g of hexadecyltrimethylammonium bromide (CTAB) was dissolved in the as-prepared solution with magnetic stirring for 0.5 h, and then 8.3 g of tetraethoxysilane (TEOS, 98%) was added. A transparent lilac solution was obtained after vigorous stirring for another hour. The precursor solution was loaded into a syringe equipped with a 22-gauge stainless steel needle and injected at a flow rate of 1 ml h$^{-1}$ via a digital syringe pump. Positive electrical potential was applied on the metallic needle and negative electrical potential on an aluminum foil-covered metallic rotating roller. The as-spun composite fibers were aged at 110 °C overnight for further crosslinking of the inorganic framework and then calcined at 500 °C in air for 4 h to remove the organics.

**Fabrication of STNMs.** A total of 1.2 g of polyvinylpyrrolidone (M$_w$: 1,300,000), 10 g of ethanol and 2 g of acetic acid were mixed by continuously stirring for 0.5 h. Then, 1.3 g of tetrabutyltitanate and 4.7 g of TEOS were dropwise added to the above mixture, and stirred overnight at room temperature till a homogenous solution was obtained. Then, the precursor solution was loaded into a syringe equipped with a 18-gauge stainless steel needle and injected at a flow rate of 3 ml h$^{-1}$ via a digital syringe pump. Positive electrical potential was applied on the metallic needle and an aluminum foil-covered grounded metallic rotating roller was used as a collector. The as-spun composite fibers were aged at 110 °C overnight and then calcined at 600 °C in air for 2 h to remove the organics.

**Fabrication of the flat SiO$_2$–TiO$_2$ composite membranes.** A total of 0.02 g of polyvinylpyrrolidone (M$_w$: 1,300,000), 30 g of EtOH and 2 g of acetic acid were mixed by continuously stirring for 0.5 h. Then 1.3 g of tetrabutyl titanate and 4.7 g of TEOS were dropwise added to the above mixture, and stirred overnight at room temperature till a homogenous solution was obtained. Then the solution was drop coated on pre-cleaned silicon wafer. The as-coated membranes were aged at 110 °C overnight and then calcined at 600 °C in air for 2 h to remove the organics.

**Characterizations.** SEM images were recorded on JSM-6510 microscopy and JEOL FE-SEM 6700 F microscopy. TEM images were obtained on JEOL JEM-2100F. CAs were measured on the Data-Physics OCA20 machine at ambient temperature and each value was obtained by measuring five different positions. Nitrogen adsorption-desorption measurements were carried out at 77.35 K on ASAP2020 after the samples being degassed at 350 °C under vacuum for 10 h. Fourier transform infrared spectrum was recorded from 400 to 4000 cm$^{-1}$ on a Nicolet Impact 410 FTIR spectrometer. The TPD experiments were performed by using a Micromeritics AutoChem II 2920 automated chemisorption analysis unit with a thermal conductivity detector under helium flow. The separation efficiency was measured by OIL480 infrared spectrometer oil content analyzer. Optical microscopy images were taken on a CMM-55E (Leica, Germany). The mechanical properties of the membranes were measured using a testing device 410R250 (TestResources, Shakopee, MN) and five samples were tested for each stack.

**Calculation methods.** We considered six cluster models, representing water, NM, and CYH molecules on the Si–OH, and Ti–OH groups, respectively, which abound on the surface of STPNMs (Fig. 4a). The dangling bonds were all saturated by H atoms. All the density functional theory calculations were carried out using DMol3 program package in Materials Studio[29]. The exchange functional PBE

(Perdew–Burke–Ernzerhof correlation)[30] was used with the general gradient approximate (GGA). In addition, the double-numerical plus polarization functions were used as the basis sets for all atoms. Before the energy calculations, the model complexes and guest molecules were fully relaxed. The binding energy was calculated according to the following equation:

$$E_{binding} = E_{adsorbent + adsorbate} - (E_{adsorbent} + E_{adsorbate})$$

**Data availability**. The data that support the findings of this study are available from the corresponding authors on request.

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

## Acknowledgements

This work is supported by the State Basic Research Project of China (Grant No. 2014CB931802), the National Key Research and Development Program of China (Grant No. 2016YFB0701100), the National Natural Science Foundation of China (Grant Nos. 21320102001, 21621001 and 21401068), and the 111 Project (Grant No. B17020).

## Author contributions

J.Y. and L.J.: Conceived and supervised the project. Y.W. and J.D.: Designed the experiments and contributed equally to this work. Y.W.: Carried out all the experiments, Y.W. and B.W. performed some of the experiments. X.L.: Carried out theoretical calculations. N.W.: Was responsible for the TEM and TPD characterization. Y.W. and J.D.: Wrote the manuscript, L.W., and Y.T.: Participated in the discussion, and J.Y. and Y.T. revised the manuscript. All authors discussed the results and commented on the manuscript.

## Additional information

**Competing interests:** The authors declare no competing financial interests.

**Change history:** A correction to this article has been published and is linked from the HTML version of this paper.

