## [Peer Review File · Nature Communications]

Reviewers' comments:

Reviewer #1 (Remarks to the Author):

This manuscript presents a novel approach of using highly hydrophilic membrane as a porous matrix for the preferable infusion and permeation of the liquids with higher surface tensions (ST) in a liquid mixture with components of different STs. The idea of using liquid-infused film as a separator is simple, effective, robust and universally applicable. I also appreciate the efforts of the authors to integrate molecular level simulation to support the experimental observations.

That being said, such an idea is not completely novel. The liquid-infused-membrane based on very low-ST matrix, or the so called liquid-gated membrane, developed by the Aizenberg group in recent years, shares significant similarity in terms of the mechanism of using differential STs for selective permeation of liquids. The authors should acknowledge this previous contribution in the introduction, and elaborate on the difference or improvements made by the current contribution.

In addition, like many other studies investigating oil-water separation using filters with special wetting properties, this study only assess the separation of originally phase-separated (or immiscible) liquid mixtures. However, most practical wastewaters in industries are in the form of emulsion. (In fact, immiscible liquid mixture does not really need filters to separate, physical skimming will be quite effective). Therefore, it would be much more compelling to justify the proposed process if it can separate miscible liquid mixtures (e.g. ethanol and water, which, to be fair, is quite challenging for all processes other than distillation) or at least water/oil emulsion (which has been achieved by many studies using membranes with special wetting properties).

Minor comments:

1). The authors spent some efforts in determining the presence of the mesopores in the PFMs. Is the existence of mesopores important or required for the functionality of the membrane? Please elaborate.

2). Line 121-123, from the values in the parentheses, I think the authors meant to compare the fluxes of the PFM with two different thicknesses. (i.e. thick vs. thin, not high pressure vs. low pressure)

Reviewer #2 (Remarks to the Author):

"What are the major claims of the paper?"

The authors are claiming a "universal strategy" to regulate wetting behavior of nanoporous fibrous membranes (PFMs) by infusing them with a high surface tension (HST) liquid to repel other low surface tension (LST) liquids as long as they are immiscible with the infused liquid.

In addition, the authors are claiming that separation of immiscible liquids with smaller differences in surface tension (ST) is achievable and is in higher demand. And switching of an existing infused liquid with another infused liquid is achievable as long as the incoming liquid has a higher ST than the existing liquid.

"Are the claims novel and will they be of interest to others in the community and the wider field? If the conclusions are not original, it would be helpful if you could provide relevant references. Is the work convincing, and if not, what further evidence would be required to strengthen the conclusions? On a more subjective note, do you feel that the paper will influence thinking in the field? Please feel free to raise any further questions and concerns about the paper. We would also be grateful if you could comment on the appropriateness and validity of any statistical analysis, as well the ability of a researcher to reproduce the work, given the level of detail provided."

The work presented is a good experimental validation of the basic principles of wetting, but does not provide a “universal strategy” or a “major insight” as claimed in the paper. Moreover, the authors themselves have demonstrated aspects of this work in previous publications, namely Ref 13, 16, 19 and 24. It is difficult to see new insight in this manuscript. From a novelty and originality aspect, this work is not suitable for Nature Communications.

Further observations are mentioned below:

- Membrane-based liquid-liquid separations are an increasingly important technology with numerous applications, particularly in petrochemical processing and wastewater treatment. This particular type of separation process fundamentally relies on the manipulation of surface forces and the preferential wetting of a membrane by one of the phases.

The displacement of one liquid within a pore by another liquid has been investigated for many years. Fundamentally, the interaction between liquids and solids has been well-studied, where the resulting equilibrium state is ultimately determined by the minimization of a system’s free energy (Quere, *Annu. Rev. Mater. Res.* 2008. 38:71–99). In the case of a liquid droplet sitting upon another immiscible liquid layer, the balancing of the three surface tensions at the contact line can be used to determine the equilibrium state, as constructed by the Neumann’s triangle conditions (1894). The correlation between displacement pressure and the system material properties (pore radius, surface tension, and contact angle between the permeating interface and pore material) has also been known for nearly 100 years (Washburn, *Phys. Rev.* 17 (3): 273. 1921). Recently, further research has built upon these principles to develop porometry techniques for characterizing membranes (Sanz et al., *Desalination* 200: 195–197 (2006)), provide further understanding of the forces at play and critical material properties (Smith et al., *Soft Matter*, 9: 1772–1780 (2013)), and also enable multiphase separation with reduced fouling behaviour (Hou et al., *Nature* 519: 70–73 (2015)). This work fails to acknowledge any of the above work. The authors themselves have explored switchable ST approaches in Ref 13, 16 and 19.

- Main concern is that new scientific insight regarding infused-liquid membranes is not obvious. Explanations on why polar surface groups are beneficial needs further validation? Insight on influence of different components of surface tension (dispersive vs polar) on wettability of PFMs would make this work more comprehensive. The binding energy argument needs to be validated as the authors fail to mention that polar and dispersive components of surface tension can alter the interaction with a surface, especially one containing a high concentration of polar hydroxyl groups as is presented in this work. More rigorous experiments could have been designed to test the stated hypothesis that liquids could be simply separated according to their surface tension. For example, the challenging separation of ethylene glycol (surface tension ~ 49 mN/m with 33 mN/m dispersive and 16 mN/m polar components) from diiodomethane (surface tension ~ 51 mN/m with 49 mN/m dispersive and 2 mN/m polar components) would not only provide a scenario where the two liquids are very close in surface tension, but also in this case the liquid with higher surface tension (that should preferentially wet the membrane according to the authors) has a much lower polar component which intuitively should result in a lower binding energy.

- It appears that the work covering fabrication of the nanoporous fibrous membranes and its characterization has been presented in Ref 24. In Ref 24, PFMs are referred to as STPNMs and have been renamed in this manuscript.

- In addition, the authors have not discussed the permeability mechanism for their infused liquid PFMs. With a pore size of ~ 1 nm, the effective permeability mechanism will be diffusion. It would be beneficial for the readers if the authors can draw a comparison with Supported Liquid Membranes (Kemperman et al., *Separation Sci. & Tech.*, Vol 31, Iss 20, 1996) that relies on nano/micro porous membranes having a stabilized infused liquid to act as part of the filter media. Comparison with these types of approaches is missing from this paper and does not provide

enough scientific insight to influence thinking in this field.

Reviewer #3 (Remarks to the Author):

This manuscript reports a new concept and protocol to separate immiscible liquids by infusing the higher-surface-tension liquid into the superamphiphilic membrane prior to use. It is interesting that the lower-surface-tension liquid infused in the membrane can be displaced by that with higher surface tension. The principle of this success relies on the different affinity of the hydrophilic surface to liquids with different surface tension. This reviewer agrees that this work targets a less-concerned topic regarding the separation of liquids with surface tension of small differences, while the authors have largely achieved this goal by using a facile protocol. The concept of this work is creative, the results are solid, and the manuscript is well organized and written. This reviewer recommends the publication of this manuscript in *Nature Communications* after minor revisions as described below.

1. It is envisioned that the very high surface energy of the as-developed material will drive the transition of its surface chemistry from Ti-OH/Si-OH into Ti-O-Ti/Si-O-Si, which will make the surface of the membrane less hydrophilic or even hydrophobic. The contamination from air onto the membrane with so high surface energy is also inevitable and thus contributes to the hydrophilicity decrease. Accordingly, how about the stability of the surface hydrophilicity and separation capability of the membrane after being stored in air for a long time?
2. Does the mesoporous structure of the material contribute to the separation efficiency of the membrane?
3. The displacement of the lower-surface-tension liquid in the membrane by that with higher surface tension should be quantitatively characterized by X-ray photoelectron spectroscopy.
4. Regarding the mechanism of this work, superhydrophilicity of the material is the key factor to ensure the success of the liquids separation. However, the superhydrophilicity of a material can be purely derived from its surface chemistry (e.g., Si-OH or Ti-OH), and it can also result from the rough and porous structure (though the material is not intrinsically superhydrophilic). This reviewer requests the authors to clarify which parameter is the key to ensure the separation of liquids with different surface tension by using the proposed protocol. In this aspect, the same tests as those conducted in this work need to be done and compared on a rough and porous substrate that shows apparent superhydrophilicity but is not made of superhydrophilic material.
5. This reviewer requests to cite the paper (K. He et al. *ACS Nano* 2015, 9, 9188–9198.) that describes a self-cleaning oil/water separation membrane that works relying on the displacement of oil by water on the superhydrophilic membrane.

Response to the reviewer's comments

Reviewer #1

This manuscript presents a novel approach of using highly hydrophilic membrane as a porous matrix for the preferable infusion and permeation of the liquids with higher surface tensions (ST) in a liquid mixture with components of different STs. The idea of using liquid-infused film as a separator is simple, effective, robust and universally applicable. I also appreciate the efforts of the authors to integrate molecular level simulation to support the experimental observations.

Reply: Thanks very much for the valuable comments and suggestions. We have provided a detailed point-to-point response to each comment.

Q1. That being said, such an idea is not completely novel. The liquid-infused-membrane based on very low-ST matrix, or the so called liquid-gated membrane, developed by the Aizenberg group in recent years, shares significant similarity in terms of the mechanism of using differential STs for selective permeation of liquids. The authors should acknowledge this previous contribution in the introduction, and elaborate on the difference or improvements made by the current contribution.

Reply: Thanks for the valuable comment. In the revised paper, we have acknowledged the previous contribution by Prof. Aizenberg in the introduction, and elaborated on the difference as well as improvements made by our current contribution.

As reported by Prof. Aizenberg's work (Nature, 2015, 519, 70, Ref 15), a low surface energy, lubricating liquid was infused in porous substrate with low surface tension and stabilized by capillary force. According to the different extra pressure of air-water-oil that is needed to open the sealed pores, the separation of a three-phase air-water-oil mixture can be realized by accurately tuning the gating threshold, named as gating mechanism (Fig. R1a). Different from their work, our separation protocol is based on the polarity-driven mechanism. A liquid with higher polar component of surface energy (PSE) was infused into the SiO₂-TiO₂ composite porous nanofibrous membranes (STPNMs, previously named as PFMs) and stabilized by the stronger interaction force with the membranes, thus realizing the repellence to the immiscible lower PSE liquid (Fig. R1b). As one component of the mixture acts as the infused-liquid, no extra pressure and additional gating liquid are required in our strategy and liquid film formed by the infused liquid will keep intact during the separation process. Furthermore, the infused liquid can be directly substituted by another immiscible liquid with much higher polarity, and then successive separation of multiphase liquid can be realized.

Figure R1. (a) Schematic illustration of the gating mechanism. For a liquid-gated-pore, flow of both gases and liquids will be gated by pressure induced deformation of the gating liquid surface. According to the different gating threshold (critical extra pressure), the separation of a three-phase air–water–oil mixture can be realized. (b) Schematic illustration of the polarity-based separation protocol of STPNMs. The STPNMs preferentially capture and interact with the high PSE liquid, forming a stable liquid-infused interface (LII), which allows the pass of infused liquid itself, but repels the immiscible liquid with lower PSE. Without any extra pressure to break the LII, the separation can be conducted solely based on gravity.

Q2. In addition, like many other studies investigating oil-water separation using filters with special wetting properties, this study only assess the separation of originally phase-separated (or immiscible) liquid mixtures. However, most practical wastewaters in industries are in the form of emulsion. (In fact, immiscible liquid mixture does not really need filters to separate, physical skimming will be quite effective). Therefore, it would be much more compelling to justify the proposed process if it can separate miscible liquid mixtures (e.g. ethanol and water, which, to be fair, is quite challenging for all processes other than distillation) or at least water/oil emulsion (which has been achieved by many studies using membranes with special wetting properties).

Reply: Thanks for the valuable comment. According to the reviewer's advice, we have successfully achieved the separation of oil-in-water emulsion (cyclohexane/water, stabilized by surfactant CTAB) and oil-in-oil emulsion (cyclohexane/formamide, stabilized by surfactant F-127) as shown in Fig. R2. For either emulsion, densely-packed droplets flood the entire view before separation process, whereas not a single droplet is observed in the collected filtrate, indicating the effectiveness of the STPNMs for separating various emulsions. However, we failed in the effective separation of miscible liquids (ethanol and water) by using STPNMs at present. As the reviewer's comment, to date, separation of miscible liquid mixtures is still a big challenge, further efforts will be made along this challenging direction.

Figure R2. The separation of emulsions by STPNMs. (a) Oil-in-water emulsion, cyclohexane/water, stabilized by surfactant CTAB. **(b)** Oil-in-oil emulsion cyclohexane/formamide, stabilized by surfactant F-127.

Q3. The authors spent some efforts in determining the presence of the mesopores in the PFMs. Is the existence of mesopores important or required for the functionality of the membrane?

Reply: The existence of mesopores in STPNMs is important for the functionality of the membrane. As shown in Fig. R3a., the STPNMs with mesopores possess higher water desorption activation energy E_d (72.35 kJ/mol) than that of the STNMs without mesopores (45.89 kJ/mol). This suggests that the water-STPNMs composite is much more stable compared to the water-STNMs composite. In addition, the stress-strain curves of STPNMs and STNMs prove that the mesopores can significantly improve the mechanical strength of membrane (Fig. R3b). Therefore the existence of mesopores in the membranes is important for achieving the high-efficient separation.

Figure R3. (a) Plots of $(2\ln T_M - \ln \beta)$ against $10^3/T_M$ for TPD of water on STPNMs and water on STPMs. The dots are experimental data measured at different temperature ramping rates and the lines are curve-fitting results. The desorption activation energies E_d are obtained by calculating the slope of fitted curve, which correspond to 72.35 kJ/mol for water on STPNMs and 45.89 kJ/mol for water on STNMs, respectively. (b) The stress-strain curves of STPNMs and STNMs. The derived Young's modulus and stress of break of STPNMs are 25.95 ± 1.9 and 2.93 ± 0.08 MPa, respectively, which are much larger than those of STNMs (2.55 ± 0.11 and 0.183 ± 0.006 MPa).

Q4. Line 121-123, from the values in the parentheses, I think the authors meant to compare the fluxes of the PFM with two different thicknesses. (i.e. thick vs. thin, not high pressure vs. low pressure)

Reply: We are sorry for unclear written in previous paper. We meant to compare the fluxes of two STPNMs with the same thickness under different extra pressures, and then draw the conclusion that the flux could be increased by increasing extra pressure.

Reviewer #2

We do appreciate the reviewer's valuable comments and extremely helpful suggestions, and believe that our revised manuscript has clearly addressed all of reviewer's concerns. In particular, according to the advice, we have thoroughly investigated the separation mechanism of our membranes, and discovered that our separation is based on the polarity-driven mechanism by which we can successfully achieve the effective separation of any immiscible liquids, or emulsion stabilized by emulsifier. Especially, we can realize the challenging separation of ethylene glycol and diiodomethane with surface energy difference only of 2 mJ/m^2 . Our polarity-driven separation protocol is universally applicable as proved by further comprehensive experimental results.

Q1. The work presented is a good experimental validation of the basic principles of wetting, but does not provide a “universal strategy” or a “major insight” as claimed in the paper. Moreover, the authors themselves have demonstrated aspects of this work in previous publications, namely Ref 13, 16, 19 and 24. It is difficult to see new insight in this manuscript. From a novelty and originality aspect, this work is not suitable for Nature Communications.

Reply: We regret that the earlier versions of our manuscript might not be written clearly enough concerning the novelty and originality aspect. So far, mainly two kinds of separation membranes are classified based on different separation mechanism. First: As illustrated in Fig. R4a, regulation of surface tension of membranes by complicated covalent modification to lie between the intrinsic wetting thresholds of the two liquids to be separated, subsequently allow the permeation of lower surface tension liquid and block the liquid with higher surface tension (such as Ref 13); Second: Membranes are superhydrophilic in air and superoleophobic underwater, which solely allows the passage of water and retains the oils (such as Ref 16, 19, 23). Previous researches on this kind of membranes are simply applied to separation of oils from water. Besides, all these referenced works utilize complicated covalent modification to alter the surfacetension of the original substrate. Ref 24 only demonstrates an adsorption film for water purification.

In our work, we firstly introduce the polarity-driven protocol for the universal separation of immiscible liquids, including not only traditional oil/water separation, but also oil/oil separation (Fig. R4b). Without any covalent modification, immiscible liquids, even with surface energy difference as small as 2 mJ/m^2 (e.g., ethylene glycol and diiodomethane), can be effectively separated. Therefore, we believe that our revised work provides major insight and new perspectives for developing new liquid separation membranes.

Figure R4. (a) Schematic illustration of the covalent modification protocol. Surface tensions (γ^{sv}) of the membrane is precisely manipulated by covalent modification to lie between the intrinsic wetting thresholds (IWTs) of the two liquids to be separated (i.e. $\gamma_{IWT}^{sv(A)} > \gamma^{sv} > \gamma_{IWT}^{sv(B)}$), so that the membrane is lyophobic for liquid A and lyophilic for liquid B. (b) Schematic illustration of the polarity-based separation protocol. The STPNMs preferentially interact with the liquid with high polar component of surface energy (PSE), forming a stable liquid-infused interface, which allows the pass of infused liquid itself, but repels the immiscible liquid with lower PSE.

Q2. Membrane-based liquid-liquid separations are an increasingly important technology with numerous applications, particularly in petrochemical processing and wastewater treatment. This particular type of separation process fundamentally relies on the manipulation of surface forces and the preferential wetting of a membrane by one of the phases.

The displacement of one liquid within a pore by another liquid has been investigated for many years. Fundamentally, the interaction between liquids and solids has been well-studied, where the resulting equilibrium state is ultimately determined by the minimization of a system's free energy (Quere, *Annu. Rev. Mater. Res.* 2008. 38:71–99). In the case of a liquid droplet sitting upon another immiscible liquid layer, the balancing of the three surface tensions at the contact line can be used to determine the equilibrium state, as constructed by the Neumann's triangle conditions (1894). The correlation between displacement pressure and the system material properties (pore radius, surface tension, and contact angle between the permeating interface and pore material) has also been known for nearly 100 years (Washburn, *Phys. Rev.* 17 (3): 273. 1921). Recently, further research has built upon these principles to develop porometry techniques for characterizing membranes (Sanz et al., *Desalination* 200: 195–197 (2006)), provide further understanding of the forces at play and critical material properties (Smith et al., *Soft Matter*, 9: 1772–1780 (2013)), and also enable multiphase separation with reduced fouling behaviour (Hou et al., *Nature* 519: 70–73 (2015)). This work fails to acknowledge any of the above work. The authors themselves have explored switchable ST approaches in Ref 13, 16 and 19.

Reply: We regret that our previous consideration was not comprehensive enough. We appreciate very much the reviewer's extremely helpful advice and have made some theoretical calculation to address the interchangeability of liquids adsorbed in our STPNMs (previously named as PFMs). We have added such relevant discussion in the revised manuscript.

Based on the fibrous substrate of our membranes, we introduce the theoretical model in terms of the minimization of a system's free energy proposed by Quere et al. (Springer, 2003, 15-18, cited as Ref 22 and Rev. Mater. Res. 2008. 38:71–99, cited as Ref 23) to study the interchangeability of liquids adsorbed in STPNMs (Fig. R5). According to the equations (1) and (2), to ensure the solid is wetted preferentially by the high polar component of surface energy (PSE) liquid (B), one should have $\Delta E_1 > 0$ and $\Delta E_2 > 0$; on the contrary, the liquid B will be substituted by low PSE liquid (A) when $\Delta E_1 < 0$ and $\Delta E_2 < 0$. In the case where only one of the conditions is satisfied, liquid B may or may not be displaced by liquid A.

$$\Delta E_1 = R(\gamma_B \cos \theta_B - \gamma_A \cos \theta_A) - \gamma_{AB} \quad (1)$$

$$\Delta E_2 = R(\gamma_B \cos \theta_B - \gamma_A \cos \theta_A) + \gamma_A - \gamma_B \quad (2)$$

Where γ_A and γ_B are the surface tensions for the liquid to be repelled and the infused liquid, γ_{AB} is the interfacial tension at the liquid–liquid interface, ϑ_A and ϑ_B are the equilibrium contact angles of the repelled liquid and the infused liquid on a flat solid surface, and R is the roughness factor. Combining with the molecular polarity listed in Supplementary Table S1, our calculating data (listed in Table R1) coincide well with the experimental results that lower PSE liquid in STPNMs can be substituted by a liquid with higher PSE.

Figure R5. The theoretical model based on the minimization of a system's free energy for maintaining a stable LII.

Table R1. Comparison of the Governing Relationships with Experimental Observations for Various STPNM-Liquid-A-Liquid-B Combinations.

Liquid A	Liquid B	R	γ_A	γ_B	γ_{AB}	ϑ_A	ϑ_B	ΔE_1	ΔE_2	Stable Film?	
										Theory	Exp.
NM	H ₂ O	2	36.8	72.8	8.32	13.7	33.6	41.45	13.77	Y	Y
H ₂ O	NM	2	72.8	36.8	8.32	33.6	13.7	-58.09	- 13.77	N	N
CYH	H ₂ O	2	25.24	72.8	48.24	5.0	33.6	22.75	23.43	Y	Y
H ₂ O	CYH	2	72.8	25.24	48.24	33.6	5.0	- 119.23	- 23.43	N	N
CYH	NM	2	25.24	36.8	5.48	5.0	13.7	15.74	9.66	Y	Y
NM	CYH	2	36.8	25.24	5.48	13.7	5.0	-26.70	-9.66	N	N
DIM	EG	2	50.8	48.8	13.85	42.6	7.0	8.24	24.09	Y	Y
EG	DIM	2	48.8	50.8	13.85	7.0	42.6	-35.94	- 24.09	N	N
DIM	NM	2	50.8	36.8	1.24	42.6	13.7	-4.52	10.72	Y/N	Y
NM	DIM	2	36.8	50.8	1.24	13.7	42.6	2.04	- 10.72	Y/N	N

Note: Satisfying both $\Delta E_1 > 0$ and $\Delta E_2 > 0$ will ensure a stable LII formation (Y). In contrast, when neither $\Delta E_1 > 0$ nor $\Delta E_2 > 0$ are satisfied, liquid B will be displaced by liquid A (N). In the case where only one of the conditions is satisfied, liquid B may or may not be displaced by liquid A (Y/N). PSE of the liquids: H₂O, 51 mJ/m²; EG, ethylene glycol, 16 mJ/m²; NM, nitromethane, 7 mJ/m²; DIM: diiodomethane, 1.8 mJ/m²; CYH: cyclohexane, 0 mJ/m².

Most of the above works suggested by the reviewer have been cited in the revised manuscript (Ref 15, 23, 24). For the liquid-gated membrane (Hou et al., Nature 519: 70–73 (2015)), an extra pressure was needed to open the pores sealed by lubricating liquid and permitted the pass of liquid. In our work, one component of the mixture acts as the infused-liquid, and no extra pressure and additional gating liquid are required and liquid film formed by the infused liquid is remains intact during the separation process.

In previous Ref 13, 16 and 19, complicated covalent modification is utilized to alter the surface of the original substrate. In this work, the wetting behaviour of STPNMs is manipulated by the infused liquid and no covalent modification is needed.

Q3. Main concern is that new scientific insight regarding infused-liquid membranes is not obvious. Explanations on why polar surface groups are beneficial needs further validation? Insight on influence of different components of surface tension (dispersive vs polar) on wettability of PFMs would make this work more comprehensive. The binding energy argument needs to be validated as the authors fail to mention that polar and dispersive components of surface tension can alter the interaction with a surface, especially one containing a high concentration of polar hydroxyl groups as is presented in this work. More rigorous experiments could have been designed to test the stated hypothesis that liquids could be simply separated according to their surface tension. For example, the challenging separation of ethylene glycol (surface tension ~ 49 mN/m with 33 mN/m dispersive and 16 mN/m polar components) from diiodomethane (surface tension ~ 51 mN/m with 49 mN/m dispersive and 2 mN/m polar components) would not only provide a scenario where the two liquids are very close in surface tension, but also in this case the liquid with higher surface tension (that should preferentially wet the membrane according to the authors) has a much lower polar component which intuitively should result in a lower binding energy.

Reply: We really appreciate the reviewer's deep insight. According to the reviewer's suggestion, we have detailedly investigated the influence of different components of surface energy of liquids on the wetting behaviour, and realized that the special wetting behaviour of STPNMs arises from a polarity-driven mechanism: the polar surface groups preferentially react with liquid with higher polar components of surface energy. Therefore, we have added new experimental data in the revised manuscript (Table. R2).

Table R2. Infused-liquid-switchable wetting behaviour of STPNMs for a series of liquid pairs.

SE (mJ/m ²)	72.8	58	48.8	36.5	44	36.8	33.3	28.4	50.8	27	25.24	25.22	20.25	18.4
Polar(mJ/m ²)	51	19	16	11.3	8	7	2.5	2.3	1.8	0.3	0	0	0	0
SP IF	H ₂ O	FM	EG	DMF	DMSO	NM	ED	TL	DIM	CCl ₄	CYH	KS	PE	NH
NH	+	+	+	+	+	+	•	•	•	•	•	•	•	•
PE	+	+	+	+	+	+	•	•	•	•	•	•	•	•
KS	+	+	+	+	+	+	•	•	•	•	•	•	•	•
CYH	+	+	+	+	+	+	•	•	•	•	•	•	•	•
CCl ₄	+	+	+	+	•	•	•	•	•	•	•	•	•	•
DIM	+	+	+	•	•	•	•	•	•	•	•	•	•	•
TL	+	+	+	•	•	•	•	•	•	•	•	•	•	•
ED	+	+	+	•	•	•	•	•	•	•	-	-	-	-
NM	+	•	+	•	•	•	•	•	-	-	-	-	-	-
DMSO	•	•	•	•	•	•	•	•	•	•	-	-	-	-
DMF	•	•	•	•	•	•	•	•	•	•	-	-	-	-
EG	•	•	•	•	•	•	•	•	•	•	-	-	-	-
FM	•	•	•	•	•	•	•	•	•	•	-	-	-	-
H ₂ O	•	•	•	•	•	•	•	•	•	•	-	-	-	-

Note: The liquids are listed according to the PSEs from high to low. The high PSE liquids infused STPNMs show lyophobic to the immiscible low PSE liquids; while the low PSE liquids infused STPNMs show lyophilic to the immiscible high PSE liquids. IF: infused; SP: separated; +: lyophobic; -: lyophilic; •: miscible; FM: formamide; DMSO: dimethylsulfoxide; DMF: N, N'-dimethylformamide; ED: ethane dichloride; TL: toluene; CCl₄: tetrachloromethane; KS: kerosene; PE: petroleum ether; NH: n-hexane.

The components of surface tension of the STPNMs are estimated based on the contact angles data and fitted by OWRK method (5 liquids were used): $\gamma_s = 57.11$ mN/m, $\gamma_s^d = 18.18$ mN/m, $\gamma_s^p = 40.93$ mN/m. The interaction of solid and liquid is the sum of the interface forces due to various types of molecular attraction containing polar and dispersive parts. As proposed by Fowkes (Attractive forces at interfaces. Ind. Eng. Chem. 56, 40-52 (1964)), the polar and dispersive interfacial attractions can be treated independently, and the polar-dispersive interactions can be neglected. For $\gamma_s^p \gg \gamma_s^d$, we can conclude that polar liquids tend to own higher interaction with STPNMs than nonpolar liquids, which equals to the special wetting behaviour.

Ethylene glycol (with higher PSE) infused membrane can repel the drop of diiodomethane (with lower PSE), whereas the ethylene glycol will permeate into the diiodomethane infused membrane (Fig. R6a,b). Thus, the challenging separation of ethylene glycol from diiodomethane (surface tension difference of 2 mN/m) can be achieved by STPNMs (Fig. R6c). Furthermore, immiscible liquids, even with polar component difference

as small as 5.2 mN/m (diiodomethane and nitromethane, shown in Fig. R6d-f), have been effectively separated.

Figure R6. Wetting behaviour and separation capability of STPNMs for EG/DIM and NM/DIM. (a,b) Photographs of contact angles of DIM droplet on EG-LII and EG droplet on DIM-LII, respectively. The EG (PSE = 16 mJ/m²) infused membrane is lyophobic to DIM, while the DIM (PSE = 1.8 mJ/m²) infused membrane is lyophilic to EG. (c) The challenging separation of EG and DIM is achieved due to the larger PSE difference. (d,e) Photographs of under NM contact angle of DIM droplet and under DIM contact angle of NM droplet, respectively. The DIM almost spreads out on the NM-LII due to the small interfacial energy difference. The NM-LII is still repellent to DIM because of the higher PSE of NM than DIM. (f) The separation of NM and DIM with NM-LII.

Q4. It appears that the work covering fabrication of the nanoporous fibrous membranes and its characterization has been presented in Ref 24. In Ref 24, PFMs are referred to as STPNMs and have been renamed in this manuscript.

Reply: We have changed PFMs to STPNMs for consistency.

Q5. In addition, the authors have not discussed the permeability mechanism for their infused liquid PFMs. With a pore size of ~1nm, the effective permeability mechanism will be diffusion. It would be beneficial for the readers if the authors can draw a comparison with Supported Liquid Membranes (Kemperman et al., Separation Sci. & Tech., Vol 31, Iss 20,

1996) that relies on nano/micro porous membranes having a stabilized infused liquid to act as part of the filter media. Comparison with these types of approaches is missing from this paper and does not provide enough scientific insight to influence thinking in this field.

Reply: Thanks for the valuable comment. The Supported Liquid Membranes (Kemperman et al., Separation Sci. & Tech., Vol 31, Iss 20, 1996) as mentioned is an ion or molecule sieving membrane, in which the adsorbed liquid allows the pass of ions or molecules by means of diffusion from feed solution to strip solution, while block the immiscible liquid (Fig. R7a). In our work, one component of the mixture acts as the infused-liquid to form a stable liquid-infused interface (LII), which can repel the immiscible liquid with lower polar component of surface energy (PSE). As the permeated and infused liquid is the same liquid, the liquid will just flow through the interfiber pores with large pore diameter (several microns) driven by gravity. Diffusion only occurs when liquid molecules permeate into the intrafiber pore (~1.7 nm), but this process has little influence on the separation flux (Fig. R7b).

Figure R7. (a) Schematic illustration of the Supported Liquid Membranes. The adsorbed liquid 2 in the supported membrane allows the pass of ions or molecules by means of diffusion, while block the immiscible liquid 1. (b) Schematic illustration of polarity-based separation protocol of STPNMs. The STPNMs preferentially capture and interact with the high PSE liquid 1, forming a stable liquid-infused interface (LII). The LII can repel the immiscible liquid 2 (red) with lower PSE while allow the permeation of the liquid 1 itself mainly from the interfiber pores.

Reviewer #3

This manuscript reports a new concept and protocol to separate immiscible liquids by infusing the higher-surface-tension liquid into the superamphiphilic membrane prior to use. It is interesting that the lower-surface-tension liquid infused in the membrane can be displaced by that with higher surface tension. The principle of this success relies on the different affinity of the hydrophilic surface to liquids with different surface tension. This reviewer agrees that this work target a less-concerned topic regarding the separation of liquids with surface tension of small differences, while the authors have largely achieved this goal by using a facile protocol. The concept of this work is creative, the results are solid, and the manuscript is well organized and written. This reviewer recommends the publication of this manuscript in Nature Communications after minor revisions as described below.

Reply: Thanks very much for the valuable comments and suggestions. We have provided a detailed point-to-point response to each comment.

Q1. It is envisioned that the very high surface energy of the as-developed material will drive the transition of its surface chemistry from Ti-OH/Si-OH into Ti-O-Ti/Si-O-Si, which will make the surface of the membrane less hydrophilic or even hydrophobic. The contamination from air onto the membrane with so high surface energy is also inevitable and thus contributes to the hydrophilicity decrease. Accordingly, how about the stability of the surface hydrophilicity and separation capability of the membrane after being stored in air for a long time?

Reply: In fact, our membranes (STPNMs, previously named as PFMs) are very stable in air for at least more than one month and retain the special wetting behaviour. Furthermore, the STPNMs can be calcined at 500 °C for regeneration if they are contaminated.

Q2. Does the mesoporous structure of the material contribute to the separation efficiency of the membrane?

Reply: The existence of mesopores in STPNMs is important for the functionality of the membrane. As shown in Fig. R8a, the STPNMs with mesopores possess higher water desorption activation energy E_d (72.35 kJ/mol) than that of the STNMs without mesopores (45.89 kJ/mol). This suggests that the water-STPNMs composite is much more stable compared to the water-STNMs composite. In addition, the stress-strain curves of STPNMs and STNMs prove that the mesopores can significantly improve the mechanical strength of membrane (Fig. R8b.). Therefore the existence of mesopores in the membranes is important for the separation efficiency.

Figure R8. (a) Plots of $(2\ln T_M - \ln \beta)$ against $10^3/T_M$ for TPD of water on STPNMs and water on STPMs. The dots are experimental data measured at different temperature ramping rates and the lines are curve-fitting results. The desorption activation energies E_d are obtained by calculating the slope of fitted curve, which correspond to 72.35 kJ/mol for water on STPNMs and 45.89 kJ/mol for water on STNMs, respectively. (b) The stress-strain curves of STPNMs and STNMs. The derived Young's modulus and stress of break of STPNMs are 25.95 ± 1.9 and 2.93 ± 0.08 MPa, respectively, which are much larger than those of STNMs (2.55 ± 0.11 and 0.183 ± 0.006 MPa).

Q3. The displacement of the lower-surface-tension liquid in the membrane by that with higher surface tension should be quantitatively characterized by X-ray photoelectron spectroscopy.

Reply: Thanks for the suggestion. FC-43 and water are selected to represent liquid with low and high polar component of surface energy, respectively (Fig. R9). Obvious characteristic peak of F 1s in condition b (red line) is observed, which is attributed to the adsorbed FC-43 in STPNMs. Subsequently, the membrane is immersed in water for several minutes and characterized by the XPS again. The peak of F 1s disappears after washing process, proving the FC-43 has been effectively substituted by water.

Figure R9. XPS spectra of STPNM for F element. (a) original STPNM, (b) FC-43 infused STPNM, (c) STPNM which was first infused by FC-43 and then washed by water.

Q4. Regarding the mechanism of this work, superhydrophilicity of the material is the key factor to ensure the success of the liquids separation. However, the superhydrophilicity of a material can be purely derived from its surface chemistry (e.g., Si-OH or Ti-OH), and it can also result from the rough and porous structure (though the material is not intrinsically superhydrophilic). This reviewer requests the authors to clarify which parameter is the key to ensure the separation of liquids with different surface tension by using the proposed protocol. In this aspect, the same tests as those conducted in this work need to be done and compared on a rough and porous substrate that shows apparent superhydrophilicity but is not made of superhydrophilic material.

Reply: The intrinsic contact angle of around 30° (Fig. R10) proves that the STPNMs is hydrophilic but not superhydrophilic materials. As reported by Tian et al. (Adv. Mater. 2016, 28, 10652), when the intrinsic contact angle of water is less than 56° , the surface should be hydrophilic under oil and oleophobic under water (hexadecane as example for oil). Combining the roughness, the wetting of surface is able to realize superhydrophilic under oil and superoleophobic under water. Therefore, both of the surface chemical composition and porous structure result in the superhydrophilicity of STPNMs and are the key to ensure the separation of liquids with different surface tension. Next, we will investigate the role of roughness and surface chemical composition in wetting behaviour of surface for immiscible liquids (not only for water and oil).

Figure R10. Photo of the intrinsic contact angle of water for STPNMs.

Q5. This reviewer requests to cite the paper (K. He et al. ACS Nano 2015, 9, 9188–9198) that describes a self-cleaning oil/water separation membrane that works relying on the displacement of oil by water on the superhydrophilic membrane.

Reply: Thanks. The reference has been cited as Ref 21.

Reviewers' comments:

Reviewer #1 (Remarks to the Author):

I think the revised manuscript is in a good shape. The introduction of the concept polar-component-of-surface-energy (PSE) is very helpful for drawing convincing conclusions. The supplementary information is very comprehensive and supportive as well. I think the manuscript is worthy of publication. I only have a few minor suggestions:

- 1). explain explicitly why PSE is used, instead of the total surface energy, as the functional parameter for determining miscibility.
- 2). On table S4, it seems like you can predict the film stability solely based on ΔE_2 . But you suggested that in theory ΔE_1 and ΔE_2 should both be higher than zero in order to have a stable film. Do you think you the condition of $\Delta E_1 > 0$ is indeed necessary?
- 3). line 75, "Preparation of STPNMS" sounds like a title in materials and methods, suggest to change to "Morphology and Properties of the STPNMS".
- 4). line 159, not clear what "for ten times" means here.

Reviewer #2 (Remarks to the Author):

The authors have re-written portions of the manuscript, and provided additional data and explanations in response to questions raised by the reviewers in the first review. The novelty of the work has been explained by emphasizing polarity-driven differences in surface energies to enable liquid-liquid separations. This mechanism is reported as an alternative to liquid-liquid separations based on manipulation of surface energies by covalent modification of membranes. The results shown and analysis performed have addressed many of the concerns raised by the reviewers; however, in my opinion, they have also raised quite a number of additional questions.

Some of the additional data and explanations that authors included in the response letter to reviewers are fundamental in understanding this work and must be incorporated in the main text or the supplementary info. For example, limitations with respect to miscible liquids or other limitations of this mechanism should be definitely discussed in the main text, even briefly. Similarly, differences with respect to "supported liquid membranes" should be mentioned or referenced in the text as well. Any limitations or impact of the pore size on the reported mechanism should be discussed.

Additional points to address are as follows:

R1Q3: With regards to STPNM comparison with STNMs for water desorption activation energy, can the authors comment on the difference in pore size between the two? STPNMs pore size has been reported as 1-2nm. What is the pore size for STNMs? Fig S5 reports STNM fiber diameter, but pore size has not been reported.

The paragraph on "separation capacity" does not add much value to the communication. In fact it is misleading. It is well known that the flux varies with membrane thickness and applied pressure. Although the section is titled "separation capacity", there is no separation occurring in these experiments (as presently described). Furthermore, the applied pressure is used nowhere else in the report, making it unclear whether this mechanism is stable under moderate to high pressures required to achieving the misleading high fluxes reported in the manuscript.

In addition, if the toluene intrusion pressure in S13a demonstrates lower PSE toluene displacing higher PSE water from the water infused STPNM, can the authors comment on this behavior in light of the polarity driven mechanism?

Main text:

Figure 1: Authors should check order of 1,2,3 in the caption (lines 364-366).

Figure 4: Minor typo on use of STPM instead of STNM in caption of 4c (line 420)

Table 1: Reason for arrows depicting IF (infused) and SP (separated) is not clearly defined.

Fabrication of the flat SiO₂-TiO₂ membrane: STPNM and STNM are discussed, but data is not discussed for (Lines 254-259)

Line 52: "Extra pressure is energy intensive" may be misleading as Ref 15 has a different feed transport mechanism compared to what is reported in this manuscript.

Line 51: Gating mechanism is not part of Ref 14

Line 98: Space missing between switchable wetting.

Line 104: SE, DSE, and PSE values should be reported for Fluorinert FC-43 (include in Table S1)

Line 119: Details of emulsion experiment should be elaborated, including relevant concentrations, volumes, and duration of separation.

Line 137: Please comment on time-scale / dynamics of the water substitution into the nitromethane infiltrated pores. Do the interfiber pores substitute first and eventually through diffusion the internal fiber pores?

Line 158: Please state the volumes of feed liquids used for cycling experiments.

Line 175: Equation is incorrect, missing γ_B .

Line 176: I appreciate the incorporation of an energy argument, but the text emphasizes the importance of the polar component of surface energy, but these equations only consider the total surface energy. Do the energy inequalities still hold if only PSE values are used? Explanations by the authors on this question would be very helpful I understanding the polarity driven mechanism.

Line 188: Typo: "angle"

S_Line 35: DST and PST should be DSE and PSE respectively.

S_F4: Y-axis should read Transmittance.

S_Line 60: Should STPNMs be STNMs?

S_Line 61: Should STNMs be STPNMs?

S_F13: Legend reads PFM.

S_Lines 191-192: Please comment on the uncertainty in these values. Is there any sample to sample variation?

Overall, the manuscript MAY become suitable for publication in Nature Communications only if the authors carefully address the above mentioned concerns and comments.

Reviewer #3 (Remarks to the Author):

In this revised manuscript, the authors have fully addressed the comments from this reviewer. I also read the authors' responses to the comments from the other reviewers, with which I am also satisfied. The additional/revised contents in the manuscript have significantly strengthened the novelty and conclusions of this work. I agree with the authors that this work opens new perspectives for the development of simple but effective strategies for liquid separation. Overall, I would recommend publication of this manuscript in Nature Communications.

Response to the reviewer's comments

Reviewer #1

I think the revised manuscript is in a good shape. The introduction of the concept polar-component-of-surface-energy (PSE) is very helpful for drawing convincing conclusions. The supplementary information is very comprehensive and supportive as well. I think the manuscript is worthy of publication. I only have a few minor suggestions:

Reply: Thanks very much. We have provided a detailed point-to-point response to each comment.

Q1. Explain explicitly why PSE is used, instead of the total surface energy, as the functional parameter for determining miscibility.

Reply: Thanks. The interaction of solid and liquid is the sum of the interface forces, including polar-polar, polar-dispersive and dispersive-dispersive interface force. Each of these interface forces can be treated independently. In our system, the calculated PSE of membranes is much larger than dispersive component of surface energy ($\gamma_S^p \gg \gamma_S^d$), thus the polar-polar interaction is dominant, and the polar-dispersive and dispersive-dispersive interactions of solid and liquid can be neglected. Therefore, the PSE is used as the functional parameter rather than the total surface energy. However, for a nonpolar substrate, such as Teflon ($\gamma_S^p \ll \gamma_S^d$), the DSE should be the key parameter. We have added this explanation in the revised paper.

Q2. On table S4, it seems like you can predict the film stability solely based on ΔE_2 . But you suggested that in theory ΔE_1 and ΔE_2 should both be higher than zero in order to have a stable film. Do you think the condition of $\Delta E_1 > 0$ is indeed necessary?

Reply: Thanks for the valuable suggestion. For the liquids listed in table S4, the stability of film can be predicted solely based on ΔE_2 , while $\Delta E_1 > 0$ is exceptional in some cases (such as nitromethane and diiodomethane). Therefore, the condition of ΔE_1 is not necessary and we have amended this in the revised manuscript (Fig. R1).

Figure R1. The theoretical model based on the minimization of a system's free energy for maintaining a stable LII.

Q3. line 75, "Preparation of STPNMS" sounds like a title in materials and methods, suggest to change to "Morphology and Properties of the STPNMS".

Reply: Thanks. We have changed the title as your suggestion.

Q4. line 159, not clear what "for ten times" means here

Reply: Thanks. Herein, we want to express that the repeating separation process of immiscible liquids on one STPNM for ten times, to test the recyclability of membrane. We have rewritten this in the revised manuscript.

Reviewer #2

The authors have re-written portions of the manuscript, and provided additional data and explanations in response to questions raised by the reviewers in the first review. The novelty of the work has been explained by emphasizing polarity-driven differences in surface energies to enable liquid-liquid separations. This mechanism is reported as an alternative to liquid-liquid separations based on manipulation of surface energies by covalent modification of membranes. The results shown and analysis performed have addressed many of the concerns raised by the reviewers; however, in my opinion, they have also raised quite a number of additional questions.

Reply: Thanks very much for the positive comments. We have provided a detailed point-to-point response to each comment.

Q1. Some of the additional data and explanations that authors included in the response letter to reviewers are fundamental in understanding this work and must be incorporated in the main text or the supplementary info. For example, limitations with respect to miscible liquids or other limitations of this mechanism should be definitely discussed in the main text, even briefly. Similarly, differences with respect to “supported liquid membranes” should be mentioned or referenced in the text as well. Any limitations or impact of the pore size on the reported mechanism should be discussed.

Reply: Thanks for the valuable suggestion. The additional data and explanations included in the previous response letter have been incorporated in the revised paper. For example, we have stated the limitation with respect to miscible liquids in the conclusion part, and the “supported liquid membranes” (Kemperman et al., Separation Sci. & Tech., Vol 31, Iss 20, 1996) has been cited as Ref 28 in the revised paper. Currently, we are not able to tune the pore size of the STPNMs for the limitation of the preparation method, and in the next step, we will comprehensively investigate the impact of the pore size on the polarity-driven liquid separation by modifying the synthetic method.

Q2. With regards to STPNM comparison with STNMs for water desorption activation energy, can the authors comment on the difference in pore size between the two? STPNMs pore size has been reported as 1-2nm. What is the pore size for STNMs? Fig S5 reports STNM fiber diameter, but pore size has not been reported.

Reply: Thanks. We have added the pore size distribution of STNMs in the revised paper. As shown in Fig. R2, the BET surface area of STPNMs reaches $652.9 \text{ m}^2 \text{ g}^{-1}$ and the pore sizes are mainly distributed between 1.2 and 2.0 nm; by contrast, STNMs possess much smaller surface area ($155.1 \text{ m}^2 \text{ g}^{-1}$) and the pore sizes distribute over a wide range from 0.5 to 20 nm. The pore sizes of both STPNM and STNM are larger than the kinetic diameter of water molecule (0.265 nm), and such pores are benefit to the diffusion of water molecules with low resistance. Therefore, we conclude that the higher water desorption activation energy

of STPNMs is mainly caused by the large surface area (about 3 times larger than that of STNMs).

Figure R2. (a) Nitrogen adsorption–desorption isotherms and (b) the corresponding pore size of STPNMs and STNMs.

Q3. The paragraph on “separation capacity” does not add much value to the communication. In fact it is misleading. It is well known that the flux varies with membrane thickness and applied pressure. Although the section is titled “separation capacity”, there is no separation occurring in these experiments (as presently described). Furthermore, the applied pressure is used nowhere else in the report, making it unclear whether this mechanism is stable under moderate to high pressures required to achieving the misleading high fluxes reported in the manuscript.

In addition, if the toluene intrusion pressure in S13a demonstrates lower PSE toluene displacing higher PSE water from the water infused STPNM, can the authors comment on this behavior in light of the polarity driven mechanism?

Reply: Thanks. We are sorry for the unclear written in the section of “separation capacity” in previous paper. The intrusion pressure is a common parameter for separation membranes (Adv. Mater. 2011, 23, 4270–4273; Adv. Mater. 2014, 26, 1771–1775), which means a critical pressure to deform the surface of the pore-filling liquid. In our work, the water layer confined in the interfiber pores will be deformed, rather than replaced by toluene when the extra pressure exceeds the intrusion pressure. The water layer will reversibly reconfigure to form a liquid-lined pore when the extra pressure decreases.

As the reviewer suggested, this part has no much relevance to the polarity driven mechanism, so we have removed it in the revised paper.

Q4. Table 1: Reason for arrows depicting IF (infused) and SP (separated) is not clearly defined.

Reply: Thanks. IL: infused liquid; RL: repellent liquid; LIM: liquid infused membrane. We have added this in the revised paper.

Q5. Fabrication of the flat SiO₂-TiO₂ membrane: STPNM and STNM are discussed, but data is not discussed for (Lines 254-259)

Reply: Thanks. We have added SEM image (Fig R3) of the flat SiO₂-TiO₂ membrane in the revised manuscript, which exhibits smooth surface.

Figure R3. SEM image of the flat SiO₂-TiO₂ membrane.

Q6. Line 52: “Extra pressure is energy intensive” may be misleading as Ref 15 has a different feed transport mechanism compared to what is reported in this manuscript.

Reply: Thanks. We have deleted this sentence in the revised manuscript.

Q7. Line 51: Gating mechanism is not part of Ref 14

Reply: Thanks. We have removed Ref 14 after gating mechanism in the revised manuscript.

Q8. Line 104: SE, DSE, and PSE values should be reported for Fluorinert FC-43 (include in Table S1)

Reply: Thanks for the valuable suggestion. We have added these values of Fluorinert FC-43 in Table S1. SE: 16.4 mJ/m²; DSE: 16.4 mJ/m²; PSE: 0 mJ/m².

Q9. Line 119: Details of emulsion experiment should be elaborated, including relevant concentrations, volumes, and duration of separation.

Reply: Thanks. The emulsions are prepared by mixing immiscible liquids with volume ratio of 1:100, and 4 mg/mL of surfactant is added under high stirring. 60 mL emulsion can be separated within 10 minutes. We have added these details in the revised paper.

Q10. Line 137: Please comment on time-scale / dynamics of the water substitution into the nitromethane infiltrated pores. Do the interfiber pores substitute first and eventually through diffusion the internal fiber pores?

Reply: Thanks. We also think the substitution process first occurs in the interfiber pores and this stage will be finished within 1 second (Fig. R4); and then water molecules will slowly

diffuse into the intra fiber pores to substitute the adsorbed nitromethane. In another work, we are investigating the dynamics of this process in detail.

Figure R4. Time-scale of the water substitution into the nitromethane infiltrated membranes. (a) Water drop (3 μL) spreads out quickly once contact with the nitromethane infused STPNMs and infiltrates into the membrane within 1 s. (b) Variation of the spreading area of a water drop (3 μL) on the nitromethane infused STPNMs.

Q11. Line 158: Please state the volumes of feed liquids used for cycling experiments.

Reply: Thanks. Feed liquids used for cycling experiments are 30 mL with volume ratio of immiscible liquids of 1:1. We have added these details in the revised paper.

Q12. Line 176: I appreciate the incorporation of an energy argument, but the text emphasizes the importance of the polar component of surface energy, but these equations only consider the total surface energy. Do the energy inequalities still hold if only PSE values are used? Explanations by the authors on this question would be very helpful in understanding the polarity driven mechanism.

Reply: Thanks for the valuable suggestion. On the base of some exceptional cases determined by ΔE_1 and according to Reviewer #1's suggestion (Question 2), the film stability can be predicted solely based on ΔE_2 and the condition of ΔE_1 is not necessary. Therefore, we have removed the condition of ΔE_1 , and ΔE_2 is denoted as ΔE . Furthermore, the PSE of STPNMs is used in the energy equation instead of total surface energy, where γ in the original expression (equation R1) is replaced by γ^p (equation R2). As shown in Table R1, the theoretical results, calculated whether with SE or PSE, agree well with the experimental results. This indicates that the energy inequalities still hold if only PSE values are used. It is because that the PSE of our membranes is much larger than DSE, and thus the polar-polar interaction between membrane and liquid is dominant. We have added these details in the revised paper.

$$\Delta E = E_A - E_B = R(\gamma_B \cos \vartheta_B - \gamma_A \cos \vartheta_A) + \gamma_A - \gamma_B > 0 \quad (R1)$$

$$\Delta E^p = E_A^p - E_B^p = R(\gamma_B^p \cos \vartheta_B - \gamma_A^p \cos \vartheta_A) + \gamma_A^p - \gamma_B^p > 0 \quad (R2)$$

Table R1. Comparison of the governing relationships with experimental observations for various STPNM-liquid-A-liquid-B combinations.

Liquid A	Liquid B	R	ΔE mJ m ⁻²	ΔE^p mJ m ⁻²	Stable Film?		
					SE	PSE	Exp.
NM	H ₂ O	2	13.77	27.36	Y	Y	Y
H ₂ O	NM	2	-13.77	-27.36	N	N	N
CYH	H ₂ O	2	23.43	33.96	Y	Y	Y
H ₂ O	CYH	2	-23.43	-33.96	N	N	N
CYH	NM	2	9.66	6.6	Y	Y	Y
NM	CYH	2	-9.66	-6.6	N	N	N
DIM	EG	2	24.09	14.91	Y	Y	Y
EG	DIM	2	-24.09	-14.91	N	N	N
DIM	NM	2	10.72	5.75	Y	Y	Y
NM	DIM	2	-10.72	-5.75	N	N	N

Note: “Y” indicates that liquid B forms a stable film, and does not get displaced by liquid A; whereas “N” indicates that Liquid B is displaced by liquid A.

Q13. S_Lines 191-192: Please comment on the uncertainty in these values. Is there any sample to sample variation?

Reply: Thanks. The temperature of desorption peaks of water, NM and CYH on STPNMs appear at 72.4 ± 0.3 °C, 66.3 ± 0.5 °C and 46.9 ± 0.4 °C, respectively, and water on STNMs at 73.8 ± 0.5 °C. The s.d. is obtained from the test results of at least three replicates. We have added these details in the revised paper.

- Figure 1: Authors should check order of 1,2,3 in the caption (lines 364-366).
- Figure 4: Minor typo on use of STPM instead of STNM in caption of 4c (line 420)
- Line 98: Space missing between switchable wetting.
- Line 175: Equation is incorrect, missing γ_B .
- Line 188: Typo: “angle”
- S_F4: Y-axis should read Transmittance.
- S_Line 35: DST and PST should be DSE and PSE respectively.
- S_Line 60: Should STPNMs be STNMs?
- S_F13: Legend reads PFM.

Reply: Thanks for the reviewer's careful review and sorry for our negligence. We have modified these mistakes in the revised paper.

Reviewer #3

In this revised manuscript, the authors have fully addressed the comments from this reviewer. I also read the authors' responses to the comments from the other reviewers, with which I am also satisfied. The additional/revised contents in the manuscript have significantly strengthened the novelty and conclusions of this work. I agree with the authors that this work opens new perspectives for the development of simple but effective strategies for liquid separation. Overall, I would recommend publication of this manuscript in Nature Communications.

Reply: Thanks very much for the positive comments.

REVIEWERS' COMMENTS:

Reviewer #2 (Remarks to the Author):

The current revision addresses all the outstanding questions, and the paper can now be considered for publication.